# Rationing Care, Job Satisfaction, Fatigue and the Level of Professional Burnout of Nurses in Urology Departments

**DOI:** 10.3390/ijerph19148625

**Published:** 2022-07-15

**Authors:** Katarzyna Jarosz, Agnieszka Zborowska, Agnieszka Młynarska

**Affiliations:** 1Department of Gerontology and Geriatric Nursing, School of Health Sciences, Medical University of Silesia, 40-635 Katowice, Poland; amlynarska@sum.edu.pl; 2Department of Clinical Nursing, Wroclaw Medical University, 50-367 Wroclaw, Poland; agnieszka.zborowska@umw.edu.pl

**Keywords:** rationing nursing care, burnout, satisfaction

## Abstract

The problem of rationing nursing care is common and present all over the world, which is a direct threat to the health and life of patients. The aim of the study was to assess the level of rationing care, fatigue, job satisfaction and occupational burnout and to assess the relationship between them and age, length of service and the number of jobs. A survey was performed among 130 Polish nurses in urology departments using the following questionnaires: Link Burnout Questionnaire, Job Satisfaction Scale, Nursing Care Rationing Scale and Modified Fatigue Impact Scale. Nursing care is rarely rationed—1.11 points; the experience of fatigue ranges between sometimes and often—52.58 points; and job satisfaction is at an average level—17.23 points. The level of rationing nursing care in urology departments is similar to that in other departments. This requires minor changes to the work of nurses to reduce the workload. Employers should develop implementation programs for young workers in order to avoid burnout and also invest in factors increasing nurses’ satisfaction, such as the atmosphere at work.

## 1. Introduction

The nursing process is the basis for maintaining the health and life of patients. Nurses, who represent a significant proportion of healthcare professionals, are responsible for its implementation. Practicing as a nurse has a significant impact on the quality of healthcare. Nurses are a professional group that, in their work, establish contact not only with the patient but also with his/her family, as well as other members of the therapeutic team [1,2]. The work of a nurse is associated with a high level of psychosocial risk, which results from the specificity of the profession, which is multi-tasking and complex [3]. The nursing profession is autonomous, and nurses are responsible for carrying out many tasks, such as nursing care, health promotion and rehabilitation. Although the nursing profession, its functions and tasks are clearly defined, the specificity of each ward affects the level of stress, occupational burnout and the level of requirements [4]. Work in the surgical ward requires from nurses: physical strength, mental resistance, constant concentration and the ability to make quick decisions [3]. The psychosocial factors influencing the work of a nurse include: high pace of work, low level of autonomy, workload, working hours, organizational culture, interpersonal relations with other employees, career development, job security, role conflict and job satisfaction [5]. Working as a nurse and helping other people requires emotional involvement. Working in an environment of human suffering has a particularly negative impact on caregivers, leading to fatigue, frustration and burnout [6]. A consequence of exposure to psychosocial factors is stress. Short periods of stress may have an adaptive function, while long-term stress may lead to somatic diseases and occupational burnout [7].

The nursing profession includes close and intense contact with other people. Involvement in work, care and responsibility for patients, constant changes, stress, working under time pressure are factors that particularly expose nurses to occupational burnout. The term was first used by the American psychiatrist Freundenberg, and then Maslach and Jackson, created a multidimensional definition, where burnout is a “psychological syndrome of emotional exhaustion, depersonalization and a reduced sense of personal achievement that can occur in people who work with other people in a certain way”. Burnout can manifest itself as a subjective symptom of high mental strain and lack of strength to work, as well as a hostile attitude towards other people and a feeling of dissatisfaction with work and having competences. The opposite issue related to professional burnout is job satisfaction, which can be defined as the relationship between investment in oneself, i.e., education, improvement of qualifications and commitment to work, and its effect, i.e., what the nurse receives, e.g., gratification, promotion and praise [8,9].

Fatigue can be characterized as a decrease in exercise capacity, which is manifested by a decrease in the intensity and efficiency of work [10]. It is associated with a sense of depressing fatigue and lack of energy, as well as a sense of lack of strength, which results from a physical and/or cognitive dysfunction of the body. In the work of a nurse, fatigue may lead to such consequences as low self-esteem, avoidance of contact with patients and many others, which may have a negative impact on the well-being and functioning in the professional and private sphere [11]. All these factors can lead to the care rationing phenomenon.

As a result of insufficient resources and pressing tasks, nurses have difficulty or are unable to complete the activities set out in the individual nursing care plan. Therefore, there are situations when they can shorten, postpone or completely abandon certain activities [12]. Rationing generally means that the trade-offs of resource scarcity are embedded in the decision-making process, with the result that necessity-driven care rationing will result in sub-optimal resource exclusion or benefit for some recipients [13]. On the other hand, rationing in healthcare can be defined as an informed and justified decision of the service provider to refuse access to medical services that extend life or medical services that may help restore or alleviate severe dysfunctions in some patients in the event of an irreversible shortage of resources. Due to this rationing assumption, medical activities are desirable and effective [14]. The attention to this problem was first raised in 2006 by the American nurse Beatrice J. Kalisch et al. [15], who created the concept of loss of nursing care, referred to as the omission error, and which refers to all areas of required patient care that have been partially or completely omitted [16]. Nursing care rationing is defined as the incomplete or non-performance of the necessary nursing activities during on-call time. It occurs when necessary care cannot be provided to patients because resources are scarce [15]. Rationing can be divided into two types, institutional and individual. Institutional rationing is manifested in the form of a specific policy of a given institution, and it is imposed on employees, e.g., nurses, doctors. On the other hand, the individual one is left for individual people and does not have specific normative foundations, rules and instructions for action. Rationing requires an individual decision in terms of metrics and ethics, which may involve providing the patient with less than optimal care. Often, the lack of involvement of the nurse in making decisions about rationing may indicate that rationing takes place at the patient’s “bedside” and that the nurse is not aware of the rationing decisions made. In the case of a deliberate rationing decision, it is burdensome for the nurse because it is morally problematic in nature [13]. Nurses’ personal confrontation with decisions about rationing care is related to the feeling of moral anxiety [17]. The reasons for rationing care include a reduction in employment, increased demand for care related to new technologies and new treatment methods, as well as the increasing level of knowledge and expectations of patients, which generates more work and time for care. The reasons for this occurrence include the selected attitudes of nurses, their knowledge and clinical evaluation during care, which may result in insufficient activities [18]. Winsett et al. identified six possible causes of the care rationing phenomenon, i.e., unexpected increase in the number of patients, increased frequency of discharges and admissions, inadequate assistants, inadequate staff, lack of availability of drugs when urgently needed and emergencies [19]. The phenomenon of rationing care is also influenced by aspects related to the nurses themselves, such as a decreased level of job satisfaction, increased level of stress, the occurrence of occupational burnout, increased absenteeism and staff turnover [20,21], in addition to factors independent of nurses, such as the work environment and culture, organizational resources, philosophy of care and model of care, as well as the financial resources allocated to the implementation of nursing care [22,23]. Prioritizing on the basis of professional clinical judgment in nursing care can be a cause of neglect as well as a negative impact on the entire therapeutic process [24].

The problem of rationing nursing care is common and present all over the world, which is a direct threat to the health and life of patients. In addition, the problems of health protection with a shortage of nurses and care and its omission are a general threat that may lead to the occurrence of medical errors [25]. The rationing of nursing care, and most of all the results, contradict the principles of holistic nursing care and lower the quality of services provided by nurses [26].

There are many studies on rationing nursing care, mostly in intensive care units or surgery units in general, for example, the research by Jankowska-Polańska et al. in the departments of hematology and pediatric oncology, where the level of rationing is high in nurses working 12 h shifts, and the level of fatigue was high in all subjects [27]. The study by Rochefort et al. in the neonatal intensive care unit showed that 28% (often) and 40% (very often) of nurses rationally prepare for discharge and provide comfort to infants [24]. Schubert’s research on surgical, gynecological and conservative departments showed that nurses rarely ignored the performance of their tasks [18]. The research of Młynarska et al. in intensive care units showed that care is rarely rationalized [11]. Studies related only to urology units are missing, which contributed to the creation of the following study. Although the nursing profession and its functions and tasks are clearly defined, the specificity of each ward affects the level of stress, occupational burnout and the level of requirements [4]. Work in the surgical ward requires such requirements as physical strength, mental resistance, constant concentration and the ability to make quick decisions [3].

Nurses working in urology departments must have detailed knowledge of the anatomy and physiology of the urogenital system of both men and women. They must have knowledge and skills enabling them to carry out the necessary procedures to care for patients in clinics and at home, performing preventive tasks, e.g., education, instruction, diagnostics (e.g., conducting uroflavometry and surgical tests, such as catheterization, suture removal), as well as caring for the sick. In urological wards, where care is mainly based on perioperative care and preparation for functioning at home. Preparing the patient to leave the hospital is a very time-consuming process that begins after the patient is admitted to the ward. Depending on the type of disease, the nurse has to educate the patient about diet, hygiene, changing stoma bags, possible side effects and the course of the perioperative process itself [28,29]. In addition, a nurse working in the urology department has contact with the most intimate sphere of a person, which requires an appropriate psychological approach to the patient and high precision during the procedures performed, as well as empathy and understanding. The ability to support the patient is of key importance, and above all, openness when talking to the patient about his intimate and sexual sphere.

Psychosocial factors cause stress in nurses, which in turn, in combination with fatigue, can affect job satisfaction and the risk of burnout, which may be transferred to the quality of patient care and the level of care rationing. The relationships between fatigue, job satisfaction, professional burnout and care rationing may be different for each nurse and of a different intensity; however, in our study, the research will be limited and averaged to the group of urological nurses.

The main aim of the study was to examine the level of rationing in nursing care, occupational burnout, fatigue and job satisfaction among nurses working in urology departments. Additionally, specific aims were set, such as: (1) the relationship between the age of the respondents and the level of care rationing, job satisfaction and professional burnout; (2) the relationship between work experience in general and work experience in the urology department and the level of care rationing, job satisfaction and professional burnout; (3) the relationship between the number of jobs and the level of job satisfaction and occupational burnout; and (4) the relationship between the level of fatigue and care rationing, the level of job satisfaction and occupational burnout.

## 2. Materials and Methods

### 2.1. Study Design

The presented study is a cross-sectional study. The research sample was 130 nurses working in urology departments from all over Poland. Data for the study were collected over a period of 3 months from March to May 2021.

According to the Central Statistical Office, in 2020 in Poland there were 160 urology wards with 3147 beds, where 188,831 patients were treated. On the other hand, in total, there were 7190 hospital wards in Poland, containing 167,567 beds, and 6,293,576 patients were treated [30]. The number and size of urological departments in relation to the general number is small, which narrows the field of urology, which means that there are fewer specialists (doctors and nurses) dealing with it than in other fields of medicine and that it is a very specialized field. The study group (130 nurses) is a representative group in its specialty.

### 2.2. Participants

A total of 130 nurses from Poland working in urology departments participated in the study. The inclusion criteria were the profession of a nurse and employment in the urology department for a minimum of 6 months, irrespective or the workload (full-time/part-time). The exclusion criteria from the study were the lack of consent to participate and incomplete completion of the questionnaire.

Most of the respondents were women (98.5%), and the average age of the respondents was 37.78 years (±11.86 years), the respondents had an average work experience of 13.31 years (±12.63 years), while the internship in the urology department was 7.71 years (±10.29 years) on average. Most respondents had a bachelor’s degree (57.7%), while almost half (46%) did not have additional postgraduate training. A significant part of the respondents had one job (61%) and worked shifts in a 12 h system (83%). Details are presented in Table 1.

### 2.3. Tools

An anonymous questionnaire was used to conduct the study, consisting of records and health questions and then several standardized questionnaires: Nursing Care Rationing Scale (PIRNCA) in the Polish adaptation of Uchmanowicz et al. [31], the Polish version of the Modified Fatigue Impact Scale (MFIS) validated and adapted by Gruszczak [32,33], the Polish adaptation of the Work Satisfaction Scale (SSP) by Zalewska [34] and the Polish adaptation of the Italian Burnout Scale (LBQ) by Santinello by Jaworowska [35]. The use of the above questionnaires allowed the examination of the care rationing, occupational burnout, fatigue and job satisfaction of nurses. All of them have been adapted to Polish conditions and validated. The paper versions of the questionnaires were used.

Rationing nursing care was first described by Shubert in 2007, who created the Basel Extent Rationing of Nursing Care (BERNCA) questionnaire to measure it [36]. In 2014, Jones in the USA adapted this questionnaire and created their own version, the PIRNCA questionnaire [37]. The PIRNCA questionnaire was used to measure the main variable, i.e., nursing care rationing. It consists of three elements. The first includes 31 questions related to care rationing, the second is the question of the quality of nursing care and the last is the question of job satisfaction. Each of the 31 questions is accompanied by a 4-point scale according to which the respondents evaluated the question: 0—never; 1—rarely; 2—sometimes; 3—often. Attached to this scale is also the answer “not applicable”. The final score is the average of the selected answers (excluding the answer “not applicable”). The final result is in the range of 0–3, and its interpretation corresponds to the scale: 0—never; 1—rarely; 2—sometimes; 3—often. In questions concerning the quality of care and satisfaction with job, a 10-point rating scale was used, where a higher rating meant higher quality of care or job satisfaction. Respondents could give 0–10 points on average for each of these questions. The PIRNCA questionnaire is a tool with a high level of reliability and validity, and the version translated into Polish is completely comparable to the original [31].

The Modified Fatigue Impact Scale (MFIS) was created on the basis of the 40-point FIS (Fatigue Impact Scale), which in turn was developed to assess fatigue among patients with chronic diseases, especially multiple sclerosis [38]. MFIS scale consists of 21 statements that concern three areas of the subject’s life: physical, cognitive and psychosocial. Each of the statements has the same score from 0 to 4, and the maximum score is 84, while each of the three areas concerning the nurse’s quality of life contains a different number of statements: 9 physical items, 10 cognitive items, and 2 psychosocial items. The respondents assessed each of the 21 statements on a scale: 0—never; 1—rarely; 2—sometimes; 3—often; 4—almost always. The final score is determined by the sum of the points. The more points the test taker receives, the more fatigue affects his/her quality of life. No standards have been defined for the MFIS scale as to what score would mean the level of high fatigue [32,33].

The basis for the creation of the Work Satisfaction Scale (SSP) was the Life Satisfaction Scale (SWLS), which allows the assessment of the cognitive part of the overall satisfaction with life. The job satisfaction survey is based on 5 job-related statements, each of which is rated on a 7-point scale: 1—strongly disagree; 2—disagree; 3—rather disagree; 4—hard to say, whether I agree or disagree; 5—I rather agree; 6—I agree; 7—I strongly agree. The examined person may receive from 5 to 35 points. The higher the score, the greater the perceived job satisfaction [34].

The Link Burnout Questionnaire (LBQ) was developed in Italy by Santinello [35]. The LBQ was used to measure the assessment of occupational burnout. This questionnaire is made up of 24 statements about the work of the examined person and the feelings associated with it. A 6-point scale was used to evaluate the statements: never, rarely, once or more times a month, more or less every week, several times a week and every day. Using the LBQ, four areas were examined: psychophysical (exhaustion—energy), relationships (lack of commitment—commitment), professional competences (lack of effectiveness—effectiveness) and existential expectations (disappointment—satisfaction). The results are in the range of 6–36 points and are divided into the following ranges: 6–10 points—low result, no symptoms of occupational burnout in the respondents; 11–25 points—average result, there is a possibility of symptoms related to burnout; 26–36 points—a high result indicates the occurrence of occupational burnout at a high level [35,39].

### 2.4. Ethical Procedure

Participation in the study was voluntary and anonymous. The nurses’ consent and the request and written consent were received prior to the assessment. Approval of the Bioethical Committee of the Medical University of Silesia in Katowice (Ethical Number: PCN/CBN/0052/KB/32/22) was obtained.

### 2.5. Data Collection and Statistical Procedures

Questionnaires were made available to nurses in paper form at urology departments (with the consent of the management) in the Silesia Voivodeship. Data from completed questionnaires were collected in an Excel spreadsheet. Statistical analyses were performed using the IBM SPSS Statistics 25 program. The basic descriptive statistics, the Kolmogorov–Smirnov distribution normality test, a series of correlation analyzes with the Pearson *r* coefficient, the Student’s *t*-test for independent samples and linear regression analysis were performed with SPSS program. The level of significance in “results” chapter was considered to be *p* = 0.05.

In order to check whether the assumption about the consistency of the distributions of the measured quantitative variables with the normal distribution has been met, the first analysis of basic descriptive statistics was carried out together with the Kolmogorov–Smirnov test. The test result was statistically significant for most of the variables. This means that their distribution differs statistically significantly from the normal curve. Nevertheless, the value of skewness for all variables does not exceed the implicit absolute value of 2, which means that these distributions are not grossly asymmetric with respect to the normal curve, even if the result of the test of the normality of the distribution is statistically significant [40]. Therefore, if the other assumptions were met, parametric tests were performed.

## 3. Results

### 3.1. Basic Descriptive Statistics

The mean score on the care rationing question was 1.11 points (±0.7 points), which means it is “rarely” rationed. The quality of patient care was 6.88 points (±1.76 points), while job satisfaction was 5.95 points (±1.92 points) on a scale of 1–10 points.

On the job satisfaction scale (SSP), the average number of points was 17.23 (±6.25 points) out of 35 possible points, which gives an average of 3.45 points for each question that can be interpreted as average job satisfaction.

In the assessment of occupational burnout (LBQ scale), the highest average scores were as follows: 22.28 points (±6.01 points) occurred in psychophysical exhaustion; in the lack of involvement in relationships with patients—20.02 points (±4.93 points); and in disappointment—19.66 points (±6.08 points). The lowest score occurred in the sense of lack of professional effectiveness—17.37 points (±4.49 points). All the results show an average result of occupational burnout, i.e., there is a risk of problems related to burnout. 

The average rating of fatigue (in the MFIS scale) is 52.68 points (±16.06 points) out of 84 points, which gives 2.5 points per question, so the experience of fatigue by the respondents ranges between “sometimes” and “often”. The average result of fatigue in the physical area is 24.10 points (±7.34 points) out of 36 points, which is 2.68 points per question, so the respondents experience fatigue in this dimension between “sometimes” and “often”. The average number of points of fatigue in the cognitive area was 25.46 points (±8.44 points) out of 40 points, which gives 2.55 points per question and, as before, indicates that the respondents experience fatigue in this area between “sometimes” and “often”. The mean score in the psychosocial area is 5.76 points (±2.13 points) out of 8, which is 2.88 points per question, which means fatigued is most often experienced between ‘sometimes’ and ‘often’.

### 3.2. Age of Respondents, Seniority, Fatigue, Rationing Care, Job Satisfaction and Burnout

The correlation analysis with Pearson’s *r* coefficient was used to examine the relationship between the age of the respondents and care rationing, job satisfaction and occupational burnout. The study showed statistically significant relationships in the case of two LBQ tool variables. It turns out that the age of the respondents correlates with psychophysical exhaustion and a sense of professional ineffectiveness. Both are weak negative relationships. This means that the older the respondents are, the lower their score on these scales is.

Then it was decided to check whether the overall seniority as a nurse and seniority in the urology department are related to the rationing of care, job satisfaction and the level of occupational burnout. To test the relationship, the correlation analysis with Pearson’s *r* coefficient was used again. It turns out that the overall length of service in the nursing profession is statistically significantly associated with two variables. The first is the job satisfaction scale, for which the Pearson *r* coefficient indicates a weak positive relationship. This means that the longer the respondents work in the nursing profession, the higher the level of job satisfaction they show. The second statistically significant correlation occurs in the case of a feeling of professional ineffectiveness, in the case of which it is a weak negative relationship. The negative nature of the relationship indicates that the longer the respondents work as a nurse, the lower their sense of professional ineffectiveness is. In the case of the remaining variables not listed, the result was statistically insignificant. Details are presented in Table 2.

It was also examined whether the level of fatigue of the respondents was related to their level of care rationing, job satisfaction and occupational burnout. The study showed statistically significant relationships between all variables, except for the relationship between psychophysical functions and the assessment of the quality of patient care, and all of them are relationships of low or moderate strength. The fatigue scales correlate positively with care rationing, patient care quality assessment and all dimensions of the LBQ tool. This means that the higher the level of fatigue in the respondents, the higher their score on the scales mentioned above. Negative associations with overall job satisfaction (PIRNCA) and the job satisfaction scale (SSP) show that the result on the above-mentioned scales decreases with increasing fatigue. Details are presented in the Table 2.

### 3.3. Number of Jobs, Job Satisfaction and Professional Burnout

Another relationship studied was to check whether nurses working in one place differ from nurses working in two or more places in terms of their level of job satisfaction and the level of occupational burnout. In order to test the differences, the Student’s t-test was performed for independent samples. The research result turned out to be statistically insignificant, which proves that the number of jobs does not differentiate the respondents in terms of the measured variables. Details are presented in the Table 3.

### 3.4. Detailed Influence of Age, Work Experience and Variables SSP, LBQ and MFIS Tools on the Scale of the PIRNCA Tool

The next stage of the study was to check which predictors statistically significantly affect the variability in the level of variables in the PIRNCA tool. The following variables were introduced into the model as predictors: age, work experience, number of jobs, job satisfaction scale, psychophysical exhaustion, lack of involvement in relationships with patients, sense of professional ineffectiveness, disappointment, physical functioning, cognitive functions and psychophysical functions.

The nursing care rationing scale was adopted as the first dependent (explained) variable. The study showed one statistically significant predictor: cognitive functions, explaining 16% of the variance alone. The non-standardized B coefficient of 0.03 indicates that when the level of cognitive function increases by one unit, the level of nursing care rationing will increase by 0.03 units. The β coefficient indicates a positive relationship between the predictor and the dependent variable. Details are presented in Table 4.

Another analysis was made for the dependent variable: general job satisfaction. In this case, we obtained three statistically significant predictors explaining 40% of the variance altogether. The first is disappointment: the non-standardized B coefficient of −0.13 means that when the level of disappointment increases by one unit, the level of overall job satisfaction will decrease by 0.13 units. The β coefficient indicates a negative relationship between the predictor and the dependent variable. The second statistically significant predictor is the job satisfaction scale, for which the coefficient B is 0.10. This means that when the level of the job satisfaction scale increases by one unit, the level of overall job satisfaction increases by 0.10 units. The last statistically significant predictor is age: the non-standardized coefficient B is −0.03, which means that with the increase of age by one unit, the level of the dependent variable will decrease by 0.03 units. Details are presented in Table 5.

Patient care quality was taken as the last dependent variable. In its case, the study also showed three statistically significant predictors explaining 13% of the variance altogether. The first one is the feeling of professional ineffectiveness: the non-standardized coefficient B of −0.8 means that with the increase in the level of this scale by one unit, the level of assessment of the quality of patient care will decrease by 0.8 units.

The second statistically significant predictor is the lack of involvement in relationships with patients. The non-standardized coefficient B (−0.07) proves that when the level of the above-mentioned variable will increase by one unit, the level of assessment of the quality of patient care will decrease by 0.07 units. The last statistically significant predictor is the number of jobs, negatively related to the dependent variable. Based on the analysis of the non-standardized B coefficient (−0.60), people working in two or more workplaces have lower scores on the patient care quality rating scale than people working in one place. Details are presented in Table 6.

## 4. Discussion

### 4.1. Rationing Nursing Care

In this study, respondents experienced fatigue in all of the areas studied, ranging between ‘sometimes’ and ‘often’. Care rationing was rated 1.11 points, so it is “rarely” rationed. In contrast, the quality of patient care was 6.88 points, and job satisfaction was 5.95 points, which means an average quality of care and job satisfaction.

The issue of rationing nursing care and its definition in Poland was first presented by Uchmanowicz et al. in 2018. [25]. Since then, researchers have investigated the levels of care rationing in various departments, mainly intensive care, conservative (non-surgical) and surgical. Studies in urology departments have not been carried out so far, so the presented results can only be compared with studies carried out in other departments. It seems important to study nursing care rationing in hospital wards with different specialties, which are less frequent or omitted in the course of research, in order to assess whether there is the same regularity in rationing care, or it differs from departments where surveys are carried out more frequently. The urology department is one of them where it is often overlooked during research or combined with the surgery department.

In studies on rationing nursing care in intensive care units conducted by Młynarska et al., the respondents felt tired in the range between “rarely” and “sometimes”, care was rationed “rarely” (0.81 pts), while the quality of care was assessed at 6.05 pts and job satisfaction at 7.13 pts. It has been shown that the greater the fatigue (MFIS), the more often care is rationed and the lower the job satisfaction. Age and seniority did not significantly affect care rationing. In the authors’ own research, care was also rationed “rarely”, but the respondents received a higher average score. The frequency of fatigue was also higher, and job satisfaction was lower. Both studies confirm that an increase in the level of fatigue causes an increase in the level of care rationing and a decrease in job satisfaction. However, unlike Młynarska et al., the impact of seniority on the level of job satisfaction was demonstrated [11].

International studies involving nurses from Croatia, the Czech Republic, Slovakia and Poland by Zeleníková et al. showed an average care rationing rating on a score of 1.13 to 1.92; that is, care rationing occurs between ‘rarely’ and ‘often’. These studies also found correlations between nursing care rationing, job satisfaction, patient care quality, age and seniority [41]. The research by M. Shubert et al. shows an average care rationing rating to be slightly less than “less frequently” (0.8 pts) [18]. These results are confirmed by the results of own research and the research of Młynarska et al.

In a study by Baszkiewicz at the department of pediatric hematology and oncology conducted using the BERNCA tool, the mean score for care rationing ranged between “rarely” and “sometimes”. A positive correlation was also shown between the age of the respondents and the level of care rationing, i.e., the older the nurse, the more often they rationalized care. The same relationship was demonstrated in the case of seniority: the longer the seniority, the more often the care is rationed [42]. These results are confirmed by the results of our own research and the research of Młynarska et al. [11].

### 4.2. Age of Respondents, Seniority, Fatigue, Rationing Care, Job Satisfaction and Burnout

In a study of the Slovak nurse population, no relationship was found between age and length of service in general and care rationing. On the other hand, there was a correlation between work experience in the current position and care rationing—nurses working for less than 5 years in the current position showed a lower level of care rationing. Care was rationed less often in ICUs than in other departments (mainly surgical, conservative, geriatric and others) [43]. In international studies of nurses from Turkey, the USA, Australia and Ireland, different results were shown than in the population of Slovak nurses. The authors showed that age and length of service influence the rationing of care, i.e., the older the nurse or the shorter the length of service, the more often they rationalized care [44]. This is not confirmed by our own research, which did not show any relationship between age and seniority and care rationing.

Research conducted among 614 Italian nurses employed in pediatric, oncology, general medicine, psychiatric, obstetric and neurological departments showed that 26% of respondents suffered from the risk of burnout [45]. On the other hand, research by Bartosiewicz and Januszewicz among nurses of primary healthcare and specialist outpatient care showed that mainly average results dominated, followed by low results: psychophysical exhaustion (69.8%—average result; 20.8% low), lack of involvement in professional relations (71.3%—average result, 16.1%—low), feeling of ineffectiveness (83.5%—average result; 1.6%—low) and disappointment (70%—average result; 18.8%—low). Age was a factor that particularly influenced the level of occupational burnout. Older nurses were characterized by a lower level of occupational burnout and psychophysical exhaustion, as well as a lower lack of involvement in professional relationships than younger people. On the other hand, the greater the length of service in the profession, the lower the level of disengagement in professional relationships [39]. Borkowska et al. in research on 105 nurses working in intensive care units, obtained average results for the areas of psychophysical exhaustion (20.33 pts), no involvement in relationships (19.17 pts), sense of professional ineffectiveness (15.9 pts) and disappointment (18.08 pts) [46]. In our study, the respondents obtained an average result in the assessment of occupational burnout, which corresponds to the results of Bartosiewicz et al. and Borkowska et al. Age influenced psychophysical exhaustion and the feeling of professional ineffectiveness, which means that the older the respondents, the lower the scores in the given spheres were. On the other hand, Sowińska’s research showed different results, i.e., no correlation between the duration of the internship and the sense of occupational burnout [47].

According to Ogińska-Bulik, occupational burnout depends on the age and seniority of employees, while employees who are younger in age and seniority are more susceptible to burnout [48]. This is partially confirmed by our own research, because age is also related to two areas of burnout: psychophysical exhaustion and the feeling of professional ineffectiveness. Hai-Ying Qu and Chun-Mei Wang, on the other hand, explain the higher incidence of burnout in younger workers by the fact that they more often take on more professional responsibilities, focus on maintaining high-quality services and seek solutions in the field of nursing care, in addition to their experience in management not being fully developed. It is also a period in the life of a nurse where she mainly develops her career and social position, as well as adapts to new social roles, e.g., the parent [49]. Wilczek-Rużyczka explains this phenomenon by the fact that young nurses have certain ideas that are verified after starting work, e.g., the idea of optimal cooperation with patients, including friendly relations, where, in reality, the patients behave differently. This leads to exhaustion and disappointment [50].

Wilczek-Rużyczka and Zaczyk conducted a meta-analysis of studies on occupational burnout among Polish nurses conducted with the use of the MBI (Maslach Burnout Inventory) tool, where they showed that the level of occupational burnout in Polish nurses is average. The results from the LBQ questionnaire in this study are within the range of the overall mean score, as in the studies of other authors, where most of the respondents indicated the average score of occupational burnout, i.e., the risk of symptoms [5].

The studies by Kanste et al. showed that nursing burnout is associated with a decreased level of care, patient satisfaction, more medical errors and increased infection and mortality rates [51].

Possible effects of professional burnout of nurses may be: lack of motivation, negative self-esteem, inadequate team relations, lack of time for patients and less empathy for them and feelings of powerlessness [52,53]. These elements will significantly reduce job satisfaction and may also cause more frequent rationing of care. On the other hand, the effects of occupational burnout towards the patient may be: lowering the quality of services, boredom and disregard for patients, treatment and indifference to problems and shortening contacts with patients [52,53]. These effects can increase care rationing; although the nurse will not be faced with the decision to ration, she will do so through experiencing burnout.

Pennsylvania conducted analyses on 95,000 nurses and showed that the level of patient satisfaction with care is lower in facilities where dissatisfied and burned-out nurses predominate [54]. This suggests that the less satisfied the nurses are, the more care is rationed.

Pawlik et al. conducted a job satisfaction survey among 227 nurses working in Poland and 203 nurses working in Norway using the standardized questionnaire Arbeitsbeschreibungsboben (ABB). Nurses from Norway were characterized by overall higher job and life satisfaction than nurses from Poland. Norwegian nurses assessed their job satisfaction as high, and Polish nurses as low. In their own research, the nurses assessed their job satisfaction as average. The main factors influencing job satisfaction were overtime, shift work, overload, shortage of staff, and lack of respect from colleagues. On the other hand, in this study, the nurses in the PIRNCA questionnaire in the open question about obstacles in performing nursing activities mentioned a too small number of nursing staff; lack of time; large workload with a large number of tasks to be performed, including documentation to be completed; lack of care and psychological support; and a lack of modern equipment and amenities. Although the question did not concern job satisfaction, these factors may affect the sense of job satisfaction, and partially overlap with those mentioned in the study by Pawlik et al. [55].

Bjørk, in his research on Norwegian nurses, showed that the older the nurses, the higher the job satisfaction level, while among Norwegian nurses working in dialysis centers, it was shown that nurses with longer work experience and higher age are more satisfied with their job [55]. Our own research confirmed that the longer the work experience, the greater the satisfaction. In addition, according to Gros, job satisfaction increases with age and seniority [56]. Different results in their research were presented by Schmalenberg et al. because, in their research, nurses with less than 5 years of work experience had higher job satisfaction than those working longer [57].

Kunecka et al. identified in their research the sources of the professional satisfaction of nurses in Poland: the image of the company, the atmosphere at work, activities and tasks performed at work and the workplace [58]. These areas correspond to what nurses reported in their own research and in Pawlik et al. [55].

The following factors influencing job satisfaction can be distinguished: atmosphere in the workplace, physical strain, employment stability, team relations, workplace equipment, work organization, defining roles, the possibility of raising qualifications, work stress and remuneration [59,60]. There is a convergence of these factors with psychosocial factors influencing the work of a nurse, which indicates that depending on whether they are negatively or positively perceived by the nurse, they may be a source of job satisfaction, or the lack of it, or will be predictors of occupational burnout. Hobbs et al. indicate the risk of long working hours; the risk of errors is higher in a 4 h shift than in an 8 h shift. In addition, the risk of errors increases significantly when working longer than 12 h, regardless of whether it is voluntary or planned. There is a high risk involved in making critical decisions when you are tired [57]. Research by Jankowska-Polańska et al. showed no correlation between education, age, the number of jobs, job satisfaction and care rationing. However, it has been shown that the higher the fatigue, the more often the care is rationed [27], which was confirmed by our own research. A study by Kalisch et al. also showed no correlation between age and care rationing [61].

The relationships between care rationing, burnout, job satisfaction, fatigue, seniority and age are complex, and it is difficult to define clear relationships between them. As can be seen from the above discussion, different authors obtained very different results and dependencies. It is undeniable that these relations should be constantly examined in various departments and on the international arena, which in the future will allow for the implementation of appropriate optimization and preventive measures.

### 4.3. Limitations and Strenghts of the Study

The limitations in the research were: (1) limitations in reaching urological wards due to the restrictions introduced in Poland related to the SARS-CoV-2 pandemic; (2) the risk of completing the questionnaire by nurses working in other departments, especially in the case of making the online version available in nursing groups; (3) a small sample of the test subjects; and (4) no previous studies in urology departments. The strength of this study was (1) the use of validated and adapted Polish questionnaires.

## 5. Conclusions

The obtained results encourage in-depth research in the field of rationing care, occupational burnout, fatigue and job satisfaction in urology departments and other departments with a lower frequency of occurrence. Research should also be expanded to include factors influencing care rationing, burnout and job satisfaction.

Care in urology departments is rarely rationed, i.e., on the same level as in other departments, and even in other nationalities. This is a positive phenomenon because rationing in urology departments does not show significant differences from other departments. Rare rationing of care requires the implementation of not necessarily large changes in the work of a nurse, e.g., increasing the number of nursing staff, employing medical tutors, organizing training in the use of equipment or new operating methods implemented. The introduced changes should also take into account reducing the workload of nurses to make them less tired, which would increase the perceived job satisfaction and reduce occupational burnout.

Employers should also consider youth worker support programs to reduce the possibility of burnout. In addition, employers should strive to strengthen the sense of professional satisfaction of nurses by taking care of appropriate remuneration, the work atmosphere and appropriate work organization.

## Figures and Tables

**Table 1 ijerph-19-08625-t001:** Demographic characteristics of nurses.

		%	n
Sex	Females	98.5	128
	Males	1.5	2
Place of residence	City	76.9	100
	Countryside	23.1	30
Marital status	Married	57.7	75
	Single	34.6	45
	Divorcee	6.9	9
	Widow	0.8	1
Education	Doctoral, PhD	0.8	1
	Medium	11.5	15
	Bachelor’s degree	57.7	75
	Master’s degree	30	39
Continuing professional development education	Qualification course of surgery nursing	15	20
	Specialization of surgery nursing	14	18
	Specialist course	34	44
	Continuing professional education	8	11
	Non-applicable	46	60
	More than 1	12	16
Operating mode	shift work in a 12 h system	83	108
	8 h shift work	5	7
	one shift work	12	15
Number of jobs held	one	61	79
	more than one	39	51

**Table 2 ijerph-19-08625-t002:** The relationship between fatigue, age, seniority and rationing care, job satisfaction and burnout.

	Total Score MFIS	Psychophysical Functions	Cognitive Functions	Physical Functioning	Age	Seniority as a Nurse	Seniority in the Urology Department	
				PIRNCA				
Nursing care rationing	0.41	0.33	0.41	0.38	0.03	0.06	0.09	*r*
	<0.001	<0.001	<0.001	<0.001	0.713	0.499	0.303	*p*
Overall job satisfaction	−0.28	−0.24	−0.28	−0.25	−0.08	−0.05	−0.08	*r*
	0.001	0.006	0.001	0.004	0.367	0.588	0.383	*p*
Patient care quality assessment	−0.22	−0.17	−0.24	−0.19	0.01	0.06	0.02	*r*
	0.012	0.054	0.006	0.032	0.948	0.504	0.865	*p*
				SSP				
Scale of job satisfaction	−0.41	−0.29	−0.40	−0.40	0.17	0.18	0.15	*r*
	<0.001	0.001	<0.001	<0.001	0.057	0.037	0.081	*p*
				LBQ				
Psychophysical exhaustion	0.49	0.45	0.44	0.5	−0.18	−0.10	0.01	*r*
	<0.001	<0.001	<0.001	<0.001	0.037	0.271	0.937	*p*
Lack of commitment to relationships with patients	0.43	0.35	0.41	0.41	−0.14	−0.13	−0.09	*r*
	<0,001	<0,001	<0,001	<0,001	0,105	0.131	0.318	*p*
Feeling of professional ineffectiveness	0.41	0.25	0.44	0.36	−0.18	−0.18	−0.05	*r*
	<0.001	0.005	<0.001	<0.001	0.04	0.036	0.555	*p*
Disappointment	0.42	0.39	0.36	0.43	−0.06	−0.02	0.3	*r*
	<0.001	<0.001	<0.001	<0.001	0.522	0.736	0.821	*p*

Abbreviation: PIRNCA—Perceived Implicit Rationing of Nursing Care; SSP—The Satisfaction With Job Scale; LBQ—Link Burnout Questionnaire; MFIS—Modified Fatigue Impact Scale.

**Table 3 ijerph-19-08625-t003:** Differences in job satisfaction and burnout depending on the number of jobs.

		95% CI				Two or More Jobs (n = 51)		One Place Work (n = 79)	
	**d Cohena**	**UL**	**LL**	* **p** *	**t**	**SD**	**M**	**SD**	**M**
			SSP						
Scale of job satisfaction	0.18	3.34	−1.10	0.319	1	5.92	16.55	6.45	17.67
			LBQ						
Psychophysical exhaustion	0.15	1.22	−3.06	0.396	−0.85	6.68	22.84	5.54	21.92
Lack of commitment to relationships with patients	0.19	0.82	−2.68	0.295	−1.05	4.72	20.59	5.05	19.66
Feeling of professional ineffectiveness	0.2	0.72	−2.47	0.279	−1.09	4.33	17.9	4.59	17.03
Disappointment	0.13	1.38	−2.95	0.476	−0.71	6.35	20.14	5.92	19.35

Abbreviation: M—average; SD—standard deviation; SSP—The Satisfaction With Job Scale; LBQ—Link Burnout Questionnaire.

**Table 4 ijerph-19-08625-t004:** Nursing care rationing predictors.

Model		F	ΔR^2^	R^2^	t	β	SE	B
1	(Constant)	24.54 *	-	0.16	1.37		0.18	0.25
	Cognitive functions				4.95 *	41	0.01	0.03

* *p* < 0.001.

**Table 5 ijerph-19-08625-t005:** Predictors of overall job satisfaction.

Model		F	ΔR^2^	R^2^	t	β	SE	B
1	(Constant)	61.37 **	-	0.32	20.10 **		0.47	9.49
	Disappointment				−7.83 **	−0.57	0.02	−0.18
2	(Constant)	12.90 **	0.06	0.38	8.83 **		0.8	7.1
	Disappointment				−5.38 **	−0.43	0.03	−0.14
	Scale of job satisfaction				3.59 **	0.29	0.03	0.09
3	(Constant)	5.08 *	0.02	0.4	9.13 **		0.86	7.87
	Disappointment				−5.40 **	−0.42	0.03	−0.13
	Scale of job satisfaction				3.96 **	0.32	0.02	0.1
	Age				−2.25 *	−0.16	0.01	−0.03

* *p* < 0.05; ** *p* < 0.001.

**Table 6 ijerph-19-08625-t006:** Patient care quality assessment predictors.

F	ΔR^2^	R^2^	t	β	SE	B	
12.28 **	-	0.09	15.00 **		0.59	8.89	(Constant)
			−3.50 **	−0.30	0.03	−0.12	A sense of lack
							professional effectiveness
5.08 *	0.04	0.11	13.82 **		0.71	9.8	(Constant)
			−2.38 *	−0.22	0.04	−0.08	A sense of lack
							professional effectiveness
			−2.25 *	−0.20	0.03	−0.07	No commitment
							in relationships with patients
4.10 *	0.03	0.13	13.49 **		0.78	10.47	(Constant)
			−2.27 *	−0.20	0.04	−0.08	A sense of lack
							professional effectiveness
			−2.16 *	−0.19	0.03	−0.07	No commitment
							in relationships with patients
			−2.02 *	−0.17	0.3	−0.60	Number of jobs

* *p* < 0.05; ** *p* < 0.001.

## Data Availability

Not applicable.

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
