# Peer review of "Rationing Care, Job Satisfaction, Fatigue and the Level of Professional Burnout of Nurses in Urology Departments"

_ijerph, 2022, doi:10.3390/ijerph19148625_

Round 1
Reviewer 1 Report
The number of the participants is low.
There is an error regarding Analysis method
The study is not significant, should include theoretical discussiıons.
Author Response
The number of the participants is low.
The group of urology nurses are representative, specific and the are a professionals with speciallized skills. It is characterized in intorduction.
There is an error regarding Analysis method I do not unterstand.
The study is not significant, should include theoretical discussiıons.
More details are included in introduction.
Thank you for your advises and opinions.
Reviewer 2 Report
The work is not to be published as it is, it would have to undergo major changes to reach its publication

Author Response
Thank you for your advises and opinions.

Reviewer 3 Report
This is an interesting study, which explores important aspects for a high quality nursing care and for the provision of a good quality of the care of the patients from the Urology departments.
However, the article needs some revisions before being considered for publication:
Introduction
- Definition of the “nursing care rationing” must be revised
- The authors should explain the reason/s why is important to conduct a study on rationing nursing care in urology units.
Material and methods
- Clarifications on the data collection tools must be provided - please see the comments in the attached file.
- More details are needed on how the questionnaires were applied, especially as the authors mention that nurses from all over Poland were included in the study.
Conclusions are merely a repetition of the main results of the research. The authors should show the relevance of their results for improving the nursing care in Urology wards.
The authors should mention the strengths and limitations of their research
Please find more comments/suggestions in the attached file.

Author Response
Introduction
- Definition of the “nursing care rationing” must be revised. It is supplemented and revised.
- The authors should explain the reason/s why is important to conduct a study on rationing nursing care in urology units. It is supplemented.
Material and methods
- Clarifications on the data collection tools must be provided - please see the comments in the attached file. It is changed.
- More details are needed on how the questionnaires were applied, especially as the authors mention that nurses from all over Poland were included in the study. It is supplemented.
Conclusions are merely a repetition of the main results of the research. The authors should show the relevance of their results for improving the nursing care in Urology wards. It is rewritten.
The authors should mention the strengths and limitations of their research. It is supplemented at the end of discussion.
Please find more comments/suggestions in the attached file. - I have also attached the pdf file.
Thank you for your advuses and opinions.

Reviewer 4 Report
The focus of this manuscript discussion is nurse burnout. This is a long-standing issue. The phenomenon of nurse burnout may be different in different countries. This also shows the value of this manuscript. After all, however, this manuscript is still an academic article. Therefore, based on academic requirements, I submit the following review comments.
1. I suggest that authors reframe the presentation logic of the entire manuscript. In the current manuscript, it is difficult for readers to understand the source of the authors' conclusions. In addition, the use of statistical methods may also need to be reconsidered. Furthermore, the sample size is too small, which will be limited the choice of statistical methods. This is also one of the fatal wounds of this manuscript.
2. In Abstract, I offer the authors the following suggestions.
(1) The writing of "Abstract" is a descriptive statement. According to IJERPH's journal style, authors are discouraged from adopting bullet-point narratives, such as distinguishing background, methods, results, and conclusions. I suggest that authors rewrite "Abstract".
(2) In line 14-15, it is inappropriate to use abbreviations in the abstract because it would be not understood by the reader. I suggest authors to correct them.
(3) In line 15-16, the representation of parentheses is not only wrong but also unnecessary. I suggest authors to correct them.
(4) In line 16-17, the authors' description is "The results of burnout spheres have slight differences, which denote average level of professional burnout." This description makes it difficult for readers to understand what the authors mean. I suggest authors to rewrite this description.
3. In line 40-44, the authors describe as follows:
“The term was first used by the American psychiatrist Freundenberg, and then Maslach and Jackson created a multidimensional definition, where burnout is a "psychological syndrome of emotional exhaustion, depersonalization and a reduced sense of personal achievement that can occur in people who work with other people in a certain way”.”
The authors are asked to provide the source of the literature citations.
4. In the “Introduction” section, I make the following suggestions.
(1) The authors should state how nurses in the urology department differ in job characteristics from other nurses. This description can highlight the value of this research.
(2) The authors should clearly state the purpose of the research.
(3) In this manuscript, the authors lack hypotheses about the research question, and the inference process by which the hypotheses are formed. This will leave readers unable to understand the authors' approach to solute the research question.
5. In the “Materials and Methods” section, the authors divide into different sections including “Study design”, “Tools”, “Participants”, and “Data collection and statistical procedures”. In accordance with IJERPH's style of formatting, authors are invited to express the above paragraphs as subsections, such as "2.1", "2.2", "2.3", and "2.4".
6. In line 80-83, authors are requested to provide the basis for the questionnaire design, or the reference source for the questionnaire.
7. In line 85-90, why did the authors use a 4-point scale for the nursing care rationing questionnaire? Also, what percentage of respondents answered "not applicable"? Is there a need to deduct them?
8. In data statistics, the authors use the sum of the scores for each item to analyze and calculate (for example: line 97-100). Although this approach is reasonable in statistical analysis, it is customary for the vast majority of studies to use the mean analysis strategy. In order to be able to compare with other studies, I suggest that the authors can use the mean method to deal with data analysis in statistics.
9. In line 136, it is clear that "α = 0.05" is the wrong expression. It should be corrected to "p = 0.05". In addition, the authors are asked to confirm whether the significance level of Pearson r is also 0.05 in the analysis tool of SPSS. Furthermore, the authors are requested to pay attention to the expression of statistical notation. The “r” of Pearson r and “p” value should be lowercase and italicized.
10. In line 137-146, the authors use Kolmogorov-Smirnov and skewness as a test for normality. For the analysis of the normality test, I make the following suggestions.
(1) I suggest that the authors justify the use of the Kolmogorov-Smirnov test for normality.
(2) The authors are invited to add a description of the normality test for Kurtosis.
11. In 148-153 and Table 1, this paragraph and table should be classified under “Participants”. The authors are invited to move them to the "Materials and Methods" section.
12. The presentation of all tables in the authors' manuscripts needs to be corrected. I make the following suggestion.
(1) By convention, variables are arranged in the leftmost column. This expression is also the writing style required by IJERPH. The authors should correct them.
(2) The numerical representation in the table is wrong. For example: "76,9" should be corrected to "76.9". The authors are invited to make corrections for numerical representations in all tables.
13. In line 194-203, the authors list the results of the open-ended questionnaire. Only 6 participants responded. This result shows that its sample size is too small in the open-ended questionnaire. As far as quantitative analysis is concerned, the significance of these answers is limited. I suggest authors can delete them. However, authors can still describe respondents' thoughts in the "Discussion" section.
14. In line 204-206, the sum of the SSP scores makes it difficult for readers to judge these values. I suggest that the authors adjust to express as an average.
15. In Table 2, this descriptive statistic is not only unnecessary but also wrong. Although descriptive statistics are also important information for quantitative analysis, authors often describe them in words or combine them in other tables based on the amount of data they provide. This avoids the confusion of manuscript themes due to lengthy statistics. Furthermore, normality must be tested for all items. There are limits to the statistical significance the authors can provide when testing constructs for normality.
16. The authors are unclear about subsection titles, including "3.4", "3.5", "3.6", "3.7", and "3.8". I suggest that the authors make appropriate corrections.
17. In Table 3 and Table 6, both tables present the results of correlation coefficients. I suggest that the authors integrate the two tables and simplify them appropriately.
18. In line 237-239, the authors are requested to revise the typography of this note.
19. The authors are invited to note the consistency of the tables and descriptions. This manuscript is missing the content of "Table 4" in the text description. The other table numbers are also wrong. The rigor of academic articles needs to be taken seriously.
20. In line 251-253, insignificant results also need to be presented.
21. In Table 7, authors are asked to justify a stepwise regression analysis strategy. In addition, in the regression analysis of categorical variables, the strategy of dummy variables must be adopted, rather than direct regression analysis. Furthermore, due to the small sample size of this manuscript, it is not appropriate to use SPSS as a tool for regression analysis. This will cause instability in the analysis results. Unless the authors use other regression analysis tools.
22. In accordance with the principle of simplicity, statistical significance is generally not described in the manuscript (e.g., line 300-302). Similar phenomena frequently appear in this manuscript. I suggest that the authors make appropriate corrections.
23. In line 326-327 and line 394-396, the authors make the following description:
“Rationing nursing care in Poland has recently been analyzed, the first in 2018 was Uchamanowicz et al.”
“Pawlik et al. conducted a job satisfaction survey among 227 nurses working in Poland and 203 nurses working in Norway using the standardized questionnaire Arbeitsbeschreibungsboben (ABB).”
The authors are requested to provide sources of citations.
24. In line 394-412, this description appears to be merely an accumulation of literature results. I don't understand how the authors want to express research inspiration for readers. Similar phenomena often appear in the descriptions of the "Discussion" section. In other words, I suggest that the authors have an in-depth discussion of the research questions and findings.
25. In line 413-415, I cannot understand the significance of the authors' analysis of job burnout in different workplaces, as it was not discussed in the literature review. In addition, the authors also lack a hypothesis on this issue.
26. In the “References” section, the authors are invited to revise the expression of references in the style of IJERPH.
Author Response

(The authors gave the same response as above.)

Reviewer 5 Report
Thank you very much for giving me the opportunity of reviewing the manuscript titled Rationing caret, job satisfaction and the level of professional burnout of nurses in urology departments. The issue is extremely important as it affects the quality of care and the nurses’ mental health and wellbeing. Reasons for rationing care should be identified and addressed.
I would suggest the following changes:
Title: it should include all the variables; fatigue is not mentioned though.
Abstract:
Please, include the aim of the study.
Some numerical results should be mentioned.
Conclusions are more results, and they should be conclusions.
Introduction
It is scarce in references, please, add more evidence supporting the arguments exposed.
Line 29: please expand it and mention the psychological risks.
Line 71: “there are many studies” please, reference these studies.
Lines 58-70: The concept of nursing care rationing is not defined in depth. This concept is quite new so it should be described with more details.
The link between rationing care and burnout, fatigue and job satisfaction are not explained. The concepts are defined one after the other, but why are you studying them together? What is their relationship?
You must mention the aim of the study at the end of the introduction.
Materials and methods:
After the study design, you should follow with the study population, sample and participants.
What is the study population? Polish nurses working in an urology ward? How many are they? Is 130 a representative number? How did you calculate the sample size?
What was the sample procedure? Was it randomized? How did you select the participants?
Please describe the procedure for data collection, how the questionnaires were distributed? were all of them valid? What was the response rate?...
Please, mention the variables before describing the tools.
When describing the tools, please, provide the reference for the original version of the scale.
Line 93, please include the score range.
Are the MFIS and work satisfaction scale validated for polish population? Please, mention it and reference.
Include ant ethical issues section
Results
Lines 148-150: please provide a SD for these means.
Section 3.1 and 3.2 are confusing. In the methods section it was not mentioned that these variables were going to be measured. They do not add any substantial information to the study, so I recommend deleting these sections.
Lines 207-221: the data provided in these lines is duplicated in table 2. I suggest summarizing them.
Table 2: I suggest eliminating D and p columns.
Tables: Please, check on all the tables. Table 4,5,6,7 and 8 do not correspond to the text.
Lines 261-262: non statistically significant.
There are too many tables, consider joining tables 3 and 5.
Discussion
This section should start with a summary of the main findings of the present study.
Lines 326-328: please provide references.
The whole discussion is written with just 8 references. This is very poor, even if the issue has not been studied or particularly in urology wards.
The discussion is not well written. Previous studies are described one after another, without relating them to each other, nor to the results of the present study. The studies are simply mentioned and described, but not interrelated. There is no defined plot line. Discussing is not only comparing your results with previous studies, but also finding a possible explanation for them, presenting a theory that supports them...
You should include the limitations for the study.
Conclusions:
This section needs to be reconsidered. I do not expect a summarize of the results, but real conclusions. Ideas, suggestions, or arguments that are coming from your results. It should also include the implication for clinical practice and future lines of investigations.
References:
Numbers are duplicated
Language and writing:
Although English is not my first language, I realize a strong need for English editing.
The study has a good base, the work and effort made is recognized, I am afraid that I cannot recommend it for publication in its current state, but it can be improved. I encourage you to follow the recommendations.
Author Response
Thank you for your advises and opinions.
Please see the attachement.

Round 2
Reviewer 2 Report
Dear editor With the changes made by the authors, the article could be published.
Author Response
Dear Reviewer, thank you very much for approving the work and our corrections.

Reviewer 3 Report
I thank the authors for carefully reviewing their manuscript. In the current version, the manuscript is much improved compared to the previous one.
However, there are still some issues that need to be corrected / clarified by the authors.
L 133- As shown in my previous review, nursing care rationing is defined by reference to the entire duration of the working time of the nurses, not only to the on-call time. Please see https://www.ncbi.nlm.nih.gov/pmc/articles/PMC6572194/
L 161- “The reasons for this phenomenon”- it is not clear the phenomenon the authors refer to
L 191- “[…] the results resulting”- please avoid repetition
L 197- “for example” is written twice
L 203- “they” could be deleted
L 204- “the research of MÅ‚ynarska et al. in intensive care units I have shown”- please reformulate
L 55-57 and L 223-225- paragraphs are identical- please delete one of them
L 334-336: “An anonymous questionnaire was used to conduct the study, consisting of a record record and health-related questions, and then one after another questionnaires that were signed”.
- This sentence seems to contain contradictory information, i.e., first the authors state that “An anonymous questionnaire was used […]” and then, at the end of the sentence mention “questionnaires that were signed”- Please revise.
- Please avoid repeating “record”.
L 343: The questionnaires were conducted by paper versions and electronic versions- the authors should detail the way in which research was conducted using the questionnaires in electronic version. This part of the research, i.e., electronic data collection was not mentioned in the previous version of the manuscript.
L 356-357: “She was the first to describe the Shubert care rationing phenomenon in 2007 and created the Basel Extent Rationing of Nursing Care (BERNCA) questionnaire to measure it [36]” - Authors should mention the name of the person they are referring to.
L 420- “conduct it” should be replaced with “participate”
L 503-504- “[…] which gives average 3.45 points for the question […]”- it seems that “the” should be replaced with “each”.
L 508- “lowest points” should be replaced with “lowest score”
L 552- “study” should be replaced with “analysis”
L 641- 645- This paragraph is not written in English, therefore I do not understand it.
L 740- ”people” should be replaced with “nurses”
L 745- “W badaniach wÅ‚asnych the respondents obtained an average result in”- Please revise, it is not clear as the beginning of the sentence is not written in English
L 760-761- The sentence “This is partially confirmed in our own research, because in two areas of burnout: psychophysical exhaustion and the feeling of professional ineffectiveness“ should be revised
L 765- the fragment “their managerial skills do not they are fully formed” should be reformulated
L 767- please check if the word “parent” is correctly used. Maybe it should be replaced with “patient”
L 793- the authors should add the bibliographical reference/s at the end of the paragraph
L 955- the word “place” should be deleted or included in the sentence by reformulating it.
In addition to those mentioned above, the entire text must be checked by the authors for other possible typos.
Author Response
Dear Reviewer,
thank you very much for your comments. All changes are marked in red in the text.
L 133- As shown in my previous review, nursing care rationing is defined by reference to the entire duration of the working time of the nurses, not only to the on-call time. Please see https://www.ncbi.nlm.nih.gov/pmc/articles/PMC6572194/
It was corrected
L 161- “The reasons for this phenomenon”- it is not clear the phenomenon the authors refer to
It was corrected
L 191- “[…] the results resulting”- please avoid repetition
It was corrected
L 197- “for example” is written twice
It was corrected
L 203- “they” could be deleted
It was corrected
L 204- “the research of MÅ‚ynarska et al. in intensive care units I have shown”- please reformulate
It was corrected
L 55-57 and L 223-225- paragraphs are identical- please delete one of them
It was corrected
L 334-336: “An anonymous questionnaire was used to conduct the study, consisting of a record record and health-related questions, and then one after another questionnaires that were signed”.
- This sentence seems to contain contradictory information, i.e., first the authors state that “An anonymous questionnaire was used […]” and then, at the end of the sentence mention “questionnaires that were signed”- Please revise.
- Please avoid repeating “record”.
It was corrected
L 343: The questionnaires were conducted by paper versions and electronic versions- the authors should detail the way in which research was conducted using the questionnaires in electronic version. This part of the research, i.e., electronic data collection was not mentioned in the previous version of the manuscript.
It was corrected, only paper version was used
L 356-357: “She was the first to describe the Shubert care rationing phenomenon in 2007 and created the Basel Extent Rationing of Nursing Care (BERNCA) questionnaire to measure it [36]” - Authors should mention the name of the person they are referring to.
It was corrected
L 420- “conduct it” should be replaced with “participate”
It was corrected
L 503-504- “[…] which gives average 3.45 points for the question […]”- it seems that “the” should be replaced with “each”.
It was corrected
L 508- “lowest points” should be replaced with “lowest score”
It was corrected
L 552- “study” should be replaced with “analysis”
It was corrected
L 641- 645- This paragraph is not written in English, therefore I do not understand it.
It was corrected
L 740- ”people” should be replaced with “nurses”
It was corrected
L 745- “W badaniach wÅ‚asnych the respondents obtained an average result in”- Please revise, it is not clear as the beginning of the sentence is not written in English
It was corrected
L 760-761- The sentence “This is partially confirmed in our own research, because in two areas of burnout: psychophysical exhaustion and the feeling of professional ineffectiveness“ should be revised
It was corrected
L 765- the fragment “their managerial skills do not they are fully formed” should be reformulated
It was corrected
L 767- please check if the word “parent” is correctly used. Maybe it should be replaced with “patient”
It was ok
L 793- the authors should add the bibliographical reference/s at the end of the paragraph
It was corrected
L 955- the word “place” should be deleted or included in the sentence by reformulating it.
In addition to those mentioned above, the entire text must be checked by the authors for other possible typos
It was corrected

Reviewer 4 Report
The revised version of this manuscript, obviously, does make substantial corrections. However, the last review comments are still uncorrected. I again raise my review comments.
1. Usually the title of the article will not have a "period". I suggest that the authors remove it (Line 3).
2. The authors list the scores for each construct in the "Abstract". This approach is inappropriate. Statistics are usually not written in "Abstract" unless absolutely necessary. This reason comes from the fact that "Abstract" mainly expresses trends.
3. In line 79-83, the authors describe as follows:
“The term was first used by the American psychiatrist Freundenberg, and then Maslach and Jackson created a multidimensional definition, where burnout is a "psychological syndrome of emotional exhaustion, depersonalization and a reduced sense of personal achievement that can occur in people who work with other people in a certain way”.”
The authors are asked to provide the source of the literature citations. This comment was raised in the last comments, but the authors did not correct it.
4. The authors have added a description of the hypothesis in the “Introduction” section. However, the authors did not discuss the effect of age, years of service, and seniority on nurses' work behaviors. I suggest that authors add literature review on this.
5. By convention, variables are arranged in the leftmost column. This expression is also the writing style required by IJERPH. The authors should correct them. This comment was raised in the last comments, but the authors did not correct it.
6. The authors add many discussions in the "Discussion" section to make this section very informative. In order to make it easier for readers to read, I suggest that authors distinguish between different subsections (including the "Limitations and strenghts of the study" subsection). I remind authors to add subsection numbers when distinguishing subsections.
7. The following text expressions, I suggest that the authors can correct them.
(1) In line 164 and 704, "ie" should be corrected to "i.e.".
(2) In line 334-335, "record" is repeated writing.
(3) In line 418, "was" should be corrected to "were".
(4) In line 427, "have" should be corrected to "has".
(5) In Table 2, "N" should be expressed as "n".
(6) In Table 2, the decimal point representation is wrong. This comment was raised in the last comments, but the authors did not correct it.
(7) In line 641-645, this text is in Polish. Authors are asked to express this text in English.
(8) In line 650, the "r" is obviously redundant. Please delete it.
(9) In line 745, the description of "W badaniach własnych" is written in Polish. Authors are requested to express in English.
(10) In line 767, "eg" should be corrected to "e.g.".
(11) In line 929-930, in typesetting, this text should have a space at the beginning of the sentence.
(12) In line 984-994, the format of "Author Contributions" does not conform to IJERPH's style requirements. Authors are invited to make appropriate corrections.
(13) In line 1004-1019, "Appendix B" is redundant. Please delete it (and no "Appendix A" is written in the manuscript).

Author Response
Dear Reviewer,
thank you very much for your comments. All changes are marked in red in the text.
- Usually the title of the article will not have a "period". I suggest that the authors remove it (Line 3).
It was corrected
- The authors list the scores for each construct in the "Abstract". This approach is inappropriate. Statistics are usually not written in "Abstract" unless absolutely necessary. This reason comes from the fact that "Abstract" mainly expresses trends.
It was corrected
- In line 79-83, the authors describe as follows:
“The term was first used by the American psychiatrist Freundenberg, and then Maslach and Jackson created a multidimensional definition, where burnout is a "psychological syndrome of emotional exhaustion, depersonalization and a reduced sense of personal achievement that can occur in people who work with other people in a certain way”.”
The authors are asked to provide the source of the literature citations. This comment was raised in the last comments, but the authors did not correct it.
It was corrected, the citation has been added
- The authors have added a description of the hypothesis in the “Introduction” section. However, the authors did not discuss the effect of age, years of service, and seniority on nurses' work behaviors. I suggest that authors add literature review on this.
In part of the introduction, the hypotheses have been removed so as not to cause confusion
- By convention, variables are arranged in the leftmost column. This expression is also the writing style required by IJERPH. The authors should correct them. This comment was raised in the last comments, but the authors did not correct it.
It was corrected
- The authors add many discussions in the "Discussion" section to make this section very informative. In order to make it easier for readers to read, I suggest that authors distinguish between different subsections (including the "Limitations and strenghts of the study" subsection). I remind authors to add subsection numbers when distinguishing subsections.
Subsections has been added
- The following text expressions, I suggest that the authors can correct them.
(1) In line 164 and 704, "ie" should be corrected to "i.e.".
It was corrected
(2) In line 334-335, "record" is repeated writing.194
It was corrected
(3) In line 418, "was" should be corrected to "were".
It was corrected
(4) In line 427, "have" should be corrected to "has".
It was corrected
(5) In Table 2, "N" should be expressed as "n".
It was corrected
(6) In Table 2, the decimal point representation is wrong. This comment was raised in the last comments, but the authors did not correct it.
It was corrected
(7) In line 641-645, this text is in Polish. Authors are asked to express this text in English.
It was corrected
(8) In line 650, the "r" is obviously redundant. Please delete it.
It was corrected
(9) In line 745, the description of "W badaniach własnych" is written in Polish. Authors are requested to express in English.
It was corrected
(10) In line 767, "eg" should be corrected to "e.g.".
It was corrected
(11) In line 929-930, in typesetting, this text should have a space at the beginning of the sentence.
It was corrected
(12) In line 984-994, the format of "Author Contributions" does not conform to IJERPH's style requirements. Authors are invited to make appropriate corrections.
It was corrected
(13) In line 1004-1019, "Appendix B" is redundant. Please delete it (and no "Appendix A" is written in the manuscript).
It was corrected

Reviewer 5 Report
Thank you very much for considering my suggestions. The manuscript has improved, but I will still recommend some minor changes:
Abstract: the aim should be placed before "survey was performed..."
In the results, section 2.2 should be "Participants" and starts with "According to the Central..."
The Ethical Issues should be the last section in the results.
Please, describe the variables.
Please, translate into English the first paragraph of the discussion.
The manuscript is difficult to read, it has a lot of cross out text, and paragraph written in non English. Please, for the next review, provide a clearer version, easier to read and understand.
Author Response
Dear Reviewer,
thank you very much for your comments. All changes are marked in red in the text.
Abstract: the aim should be placed before "survey was performed..."
It was corrected
In the results, section 2.2 should be "Participants" and starts with "According to the Central..."
The Ethical Issues should be the last section in the results.
It was corrected
Please, describe the variables.
It was corrected
Please, translate into English the first paragraph of the discussion.
It was corrected
The manuscript is difficult to read, it has a lot of cross out text, and paragraph written in non English. Please, for the next review, provide a clearer version, easier to read and understand.
It was corrected
